# A PID Controller Approach for Adaptive Probability-dependent Gradient Decay in Model Calibration

**Siyuan Zhang**
School of Internet of Things Engineering
Jiangnan University
Wuxi, China 214122

**Linbo Xie** *
School of Internet of Things Engineering
Jiangnan University
Wuxi, China 214122

## Abstract

Modern deep learning models often exhibit overconfident predictions, inadequately capturing uncertainty. During model optimization, the expected calibration error tends to overfit earlier than classification accuracy, indicating distinct optimization objectives for classification error and calibration error. To ensure consistent optimization of both model accuracy and model calibration, we propose a novel method incorporating a probability-dependent gradient decay coefficient into loss function. This coefficient exhibits a strong correlation with the overall confidence level. To maintain model calibration during optimization, we utilize a proportional-integral-derivative (PID) controller to dynamically adjust this gradient decay rate, where the adjustment relies on the proposed relative calibration error feedback in each epoch, thereby preventing the model from exhibiting over-confidence or under-confidence. Within the PID control system framework, the proposed relative calibration error serves as the control system output, providing an indication of the overall confidence level, while the gradient decay rate functions as the controlled variable. Moreover, recognizing the impact of gradient amplitude of adaptive decay rates, we implement an adaptive learning rate mechanism for gradient compensation to prevent inadequate learning of over-small or over-large gradient. Empirical experiments validate the efficacy of our PID-based adaptive gradient decay rate approach, ensuring consistent optimization of model calibration and model accuracy. The code of implementation is available in https://github.com/UHIF/PID_AGD.

## 1 Inroduction

Model calibration aims to refine the uncertainty distribution of model output, ensuring its faithful reflection of the inherent uncertainty characteristics. Softmax mapping is commonly employed in the baseline learning to establish the output-probability mapping. Yet, with the escalation in model parameterizations, the output uncertainty distribution from over-parameterized models tends to become over-confident [1]. Consequently, the baseline strategy, which exclusively depends on Softmax mapping without accounting for calibration characteristics as optimization objectives, fails to achieve perfect model calibration. Particularly in high-risk applications, inadequate model calibration poses heightened safety risks [2, 3].

There are three primary strategies for calibrating uncertainty in deep learning models: Bayesian neural networks, post-processing calibration, and training-based model calibration. Bayesian neural networks (BNNs) integrate Bayesian inference into their framework, distinguishing them from traditional neural networks [4, 5]. In contrast to conventional neural networks, which yield point estimates,

---

*Linbo Xie is the corresponding author. E-mail: xie_linbo@jiangnan.edu.cn

BNNs offer a probability distribution across potential outputs, enabling uncertainty quantification in predictions [6]. This is accomplished by assigning prior distributions to the network's parameters and iteratively updating these distributions using Bayes' theorem as more data is acquired.

The post-processing calibration methods involve establishing the additional output-probability relationship through supplementary mappings to refine output uncertainty distribution [7, 8, 9]. This calibration method avoids disrupting the original model's decision-making pertaining to the primary task, maintaining the original generalization performance [10]. However, a notable drawback emerges: the necessity of additional structure to establish the mapping between outputs and probabilities [11]. This task possesses unique characteristics, demanding calibrated properties to construct output-probability mapping structures, optimization goals, and strategies. Unlike typical loss functions and metrics in machine learning, uncertainty is sample-specific; however, validating it individually is infeasible [12]. Furthermore, when validated collectively, it fails to fully represent individual sample properties, presenting challenges in calibrating individual sample. This presents a difficulty in calibrating model confidence via post-processing structures [13, 14].

Another approach to model calibration involves integrating various elements into the baseline optimization strategy for deep learning [7]. This enhancement of elements during the optimization leads to an improvement in the uncertainty distribution of the model output. These methods encompass a range of techniques including pre-training [15], data augmentation [16], label smoothing [17], weight decay [1], early stopping [18], structure sparsity [19], convolutional structure [20], and others. These methods not only bolster the calibration of the model and refine the output uncertainty distribution but also delve into the dynamic attributes of the model optimization, thereby enhancing decision-making. Additionally, they improve the interpretability in the decision-making and optimization process. Furthermore, certain loss functions for model output distribution have been devised based on uncertainty properties. These include MMCE [14], Correctness Ranking Loss [21], CALS [22], Focal loss [23, 24], and FLSD [25]. They jointly address classification accuracy and confidence calibration to mitigate the inclination of the over-confidence and under-confidence.

In the optimization of Softmax-based cross-entropy loss, model accuracy and model calibration represent distinct optimization characteristics [26]. Model calibration tends to be overfitting earlier in the optimization compared to accuracy [1]. To ensure consistent optimization between model accuracy and calibration, it is imperative to achieve high accuracy while maintaining adequate calibration. Our approach introduces a hyperparameter that controls the gradient decay rate within the Softmax output-probability mapping. It indicated a negative correlation between the gradient decay rate with increasing instance-level probability and the overall confidence distribution [27]. To achieve consistent optimization of accuracy and calibration, we propose detecting model calibration through a validation set in the optimization process. Drawing inspiration from the notable success of proportional-integral-derivative (PID) controllers in automatic control systems [28], we introduce an Adaptive Probability-dependent Gradient Decay through PID controller approach to calibrate the model. Moreover, considering that probability-dependent gradient decay rate may impact gradient amplitude, we design a dynamic learning rate mechanism corresponding to the changing gradient decay rate to offset fluctuations in gradient amplitude.

Our main contributions in this work can be summarized as follows: (1) We propose adaptive probability-dependent gradient decay via PID controller. This approach utilizes the feedback mechanism from automatic control to detect model calibration and adjust the probability-dependent gradient decay rate coefficient in Softmax, ensuring consistent optimization of model calibration and accuracy. (2) To counteract fluctuations in gradient magnitude caused by the adaptive probability-dependent gradient decay rate, we introduce a dynamic learning rate schedule to follow the dynamic decay rate. (3) Empirical experiments confirm the effectiveness of our approach, achieving improved model calibration while maintaining the model's generalization ability.

## 2 Problem formulation

### 2.1 Model calibration

Considering a dataset $\left\{(x^i, y^i)\right\}_{i=1}^{N} \subset \mathbf{R}^n \times \mathbf{R}^m$ and classifier $f$ maps $x$ to the outputs $z_j, j = 1, \ldots, m$ on $m$ classes and $k = \arg\max_j z_j$. The ground-truth $y$ and predicted labels $\hat{y}$ are formulated in one-hot format where $y_c = 1$ and $\hat{y}_k = 1$, where $c$ represents the truth class. The associated confidence score of the predicted label in baseline is $\hat{p} = \max s_j(z), j = 1, \ldots, m$, where $s(\cdot)$

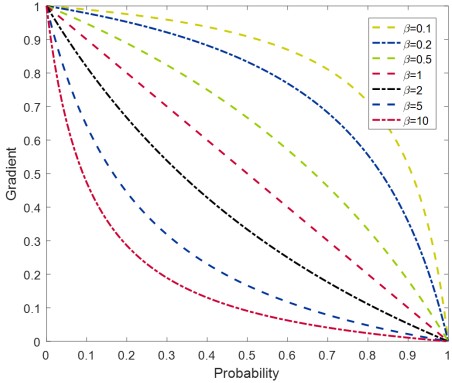

Figure 1: The gradient magnitude of different gradient decay $\beta$ with increasing probability $p_c$.

represents Softmax mapping $R^m \to R^m$. However, Softmax mapping probabilities are not accurately reflected in the properties of model output [1]. The probability estimation of modern deep models may show over-confidence or under-confidence.

**Confidence Calibration** Perfect calibration of neural network can be realized when the confidence score reflects the real probability that the classification is classified correctly. Formally, the perfectly calibrated network satisfied $\mathrm{P}\left(\hat{y} = y | \hat{p} = p\right) = p$ for all $p \in [0, 1]$. However, in practical applications, the sample is divided into $M$ bins $\{D_b\}_{b=1}^{M}$. The limited availability of data restricts the ability to accurately estimate the calibration error. According to their confidence scores and the calibration error, an approximation is calculated for each bins $\{D_b\}_{b=1}^{M}$. $D_b$ contains all sample with $\hat{p} \in \left[\frac{b}{M}, \frac{b+1}{M}\right)$. Average confidence is computed as $conf\left(D_b\right) = \frac{1}{|D_b|} \sum_{i \in D_b} \hat{p}^i$ and the bin accuracy is computed as $acc\left(D_b\right) = \frac{1}{|D_b|} \sum_{i \in D_b} \mathrm{I}\left(y_c^i = \hat{y}_c^i\right)$. ECE and MCE [29] are calculated as follows.

$$ECE = \sum_{b=1}^{M} \frac{|D_b|}{N} \left|acc\left(D_b\right) - conf\left(D_b\right)\right| \tag{1}$$

$$MCE = \max_{b \in \{1,\dots,M\}} \left|acc\left(D_b\right) - conf\left(D_b\right)\right| \tag{2}$$

### 2.2 Parametric Softmax

Softmax cross-entropy (CE) is expressed as

$$J = -\log \frac{e^{z_c}}{\sum_{j=1}^{m} e^{z_j}} \tag{3}$$

We introduce two hyperparameters in the Softmax mapping, which is expressed as follows:

$$J = -\log \frac{e^{z_c/\tau}}{\sum_{j \neq c} e^{z_j/\tau} + \beta e^{z_c/\tau}} \tag{4}$$

The parametric Softmax cross-entropy can be approximated as the following max function, as shown in (5). Minimizing this max function is expected that output $z_c$ can be larger than other class outputs $z_j, j = 1, \dots, m, j \neq c$, which is in line with the logic of the one-versus-all classification decision-making $cls\left(z\left(x\right)\right) = \max\left\{z_j\left(x\right)\right\}, j = 1, \dots, m$.

$$\lim_{\tau \to 0} -\log \frac{e^{z_c/\tau}}{\sum_{j \neq c} e^{z_j/\tau} + \beta e^{z_c/\tau}} = \lim_{\tau \to 0} \max\left\{\log \beta, z_j - z_c/\tau, j = 1, \dots, m, j \neq c\right\} \tag{5}$$

The temperature coefficient $\tau$ has been extensively investigated in model calibration. Temperature scaling represents a commonly utilized calibration technique in post-processing calibration methods. $\beta$ is regarded as a soft margin, with its approximation procedure detailed in Eq. (5). Consequently,

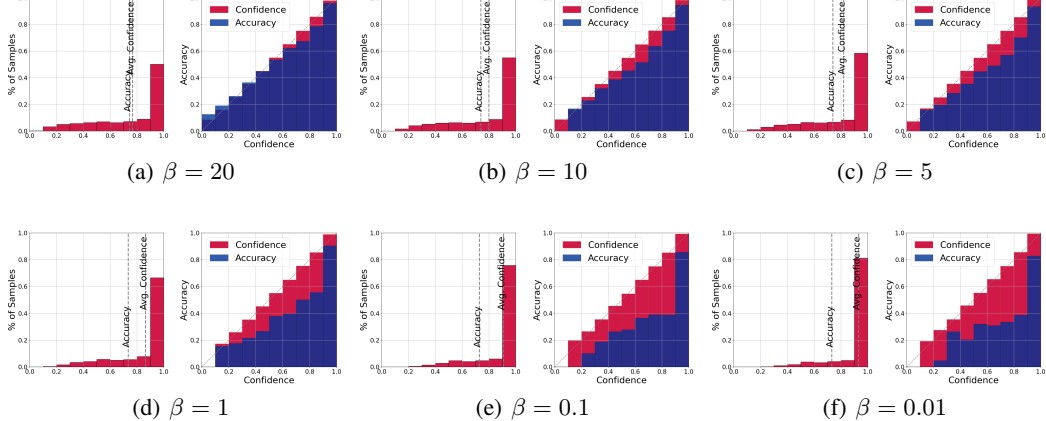

(a) $\beta = 20$    (b) $\beta = 10$    (c) $\beta = 5$

(d) $\beta = 1$    (e) $\beta = 0.1$    (f) $\beta = 0.01$

Figure 2: **Confidence and reliability diagrams with ResNet18 on CIFAR-100.** ($bins = 10$) In each subplot, the left plot illustrates the sample distribution in individual bins, while the right plot displays the average confidence and accuracy in each bin. Ideally, calibration aims for consistency between accuracy and average confidence in each bin. It indicates that a smaller gradient decay rate $\beta$ is associated with more pronounced miscalibration of the model, while a larger gradient decay rate mitigates this issue.

CE can be interpreted as a margin-based loss function. Nonetheless, due to the distance distortion between input and representation spaces, maximizing the margin in the input space of models is not achieved simultaneously by large margin Softmax. Consequently, its dynamic characteristic in the optimization process tends to be ambiguous.

### 2.3 Probability-dependent gradient decay

Considering the Softmax with the sole hyperparameter $\beta$, the temperature $\tau$ is set to 1.

$$J = -\log \frac{e^{z_c}}{\sum_{j \neq c} e^{z_j} + \beta e^{z_c}} \tag{6}$$

Let us first consider the gradient of the Softmax.

$$\frac{\partial J}{\partial z_c} = -\frac{\sum e^{z_j} - e^{z_c}}{\sum e^{z_j} + (\beta - 1) e^{z_c}} \tag{7}$$

$$\frac{\partial J}{\partial z_j} = \frac{e^{z_j}}{\sum e^{z_j} + (\beta - 1) e^{z_c}} \tag{8}$$

Introducing probabilistic output $p_j = \frac{e^{z_j}}{e^{z_1} + \cdots + e^{z_m}}$ as an intermediate variable, we obtain

$$\frac{\partial J}{\partial z_j} = \begin{cases} -\dfrac{1 - p_c}{1 + (\beta - 1)p_c}, j = c \\ \dfrac{p_j}{1 + (\beta - 1)p_c}, j \neq c \end{cases} \tag{9}$$

Fig. 1 illustrates how the introduced hyperparameter $\beta$ determines the gradient decay rate as the instance-level probability increases. More empirical experimental results can be found in the Appendix A.2. A smaller $\beta$ results in a reduced decay rate of gradient amplitude corresponding to probability. Empirical investigations have shown that the magnitude of $\beta$ during optimization determines the average confidence level and consequently impacts the calibration performance of the model. Fig. 2 demonstrates that a low gradient decay rate exacerbates the over-confidence in the model's output, whereas a high gradient decay rate can alleviate this issue and yield improved calibration results [27].

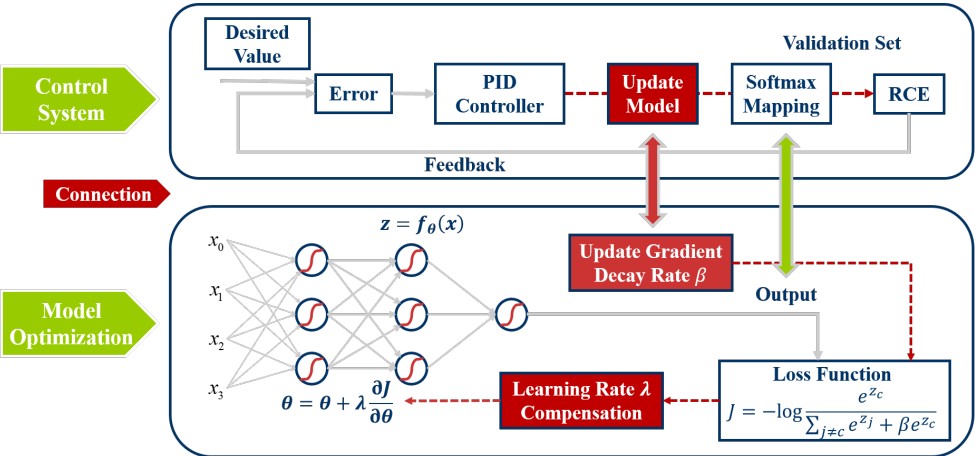

Figure 3: The framework of PID controller-based adaptive probability-dependent gradient decay.

## 3 Methodology

The probability-dependent gradient decay rate exhibits a negative correlation with the model's average confidence, as shown in Fig. 2 and Appendix A.2. Higher gradient decay rates result in a concave reduction in gradient magnitude as the sample confidence increases, thereby yielding a smoother distribution of confidence. Although high gradient decay rates can mitigate over-confidence distributions for samples, excessively small or large gradient magnitudes may lead to inadequate optimization. Consequently, we propose an adaptive gradient decay rate by a PID controller to ensure model calibration on optimization. Additionally, we propose a variable learning rate schedule to adjust the gradient and counterbalance the impact of fluctuating gradient decay rates on gradient magnitude. The whole framework is shown in Fig. 3.

### 3.1 PID controller approach for adaptive probability-dependent gradient decay

Our approach focuses on regulating the gradient decay rate by monitoring the average confidence and accuracy within each bin of the validation set throughout model optimization. This aims to enhance the model's calibration and dynamic properties during optimization. Eq. (1) specifies the desired calibration by representing only the absolute value of the difference between average confidence and accuracy within each bin. However, it fails to capture the under-confidence and over-confidence in model calibration. Therefore, we propose a Relative Calibration Error (RCE) to reflect over-confidence and under-confidence:

$$RCE = \sum_{b=1}^{M} \frac{|D_b|}{N} \left( conf\left( D_b \right) - acc\left( D_b \right) \right) \tag{10}$$

In the control system depicted in Fig. 3, RCE serves as the output, with the gradient decay coefficient acting as the controlled variable. The target RCE value is set to 0. During each iteration, the model processes the validation set to calculate the RCE. If the RCE is greater than 0, it indicates over-confidence in the probability distribution, necessitating an increase in the gradient decay rate during model optimization. Conversely, if the RCE is less than 0, it signifies under-confidence in the probability distribution, prompting a decrease in the gradient decay rate. The PID controller determines the specific adjustment required for the gradient decay coefficient $\beta$. A PID controller continually computes an error $e\left(t\right)$, representing the disparity between the desired optimal RCE and the control system output. It then applies a correction $u\left(t\right)$ to the system, incorporating proportional $(P)$, integral $(I)$, and derivative $(D)$ terms of $e\left(t\right)$. Mathematically, there is:

$$u\left(t\right) = K_p e\left(t\right) + K_i \int_0^t e\left(t\right) dt + K_d \frac{\partial}{\partial t} e\left(t\right) \tag{11}$$

**Algorithm 1:** PID controller-based adaptive gradient decay

---

**Data:** Training set $\left\{x_i^{train}, y_i^{train}\right\}_{i=1}^{N}$; Validation set $\left\{x_i^{val}, y_i^{val}\right\}_{i=1}^{N_{val}}$; Classification model $f_\theta$; Learning rate $\lambda$; Batch size $N_B$; Maximum iteration $T_{max}$; PID controller $K_p$, $K_i$ and $K_d$.

**Result:** Classification model $f_\theta$

1 Initialize $\theta$, $\beta$, $\lambda$;
2 **while** $t < T_{max}$ **do**
3     *# Adjust gradient decay rate $\beta$ by PID controller*;
4     $RCE$ of validation set $\leftarrow$ Computing by Eq.(10);
5     $u\left(t\right) \leftarrow$ Computing by Eq.(11);
6     $\beta_t \leftarrow$ Computing by Eq.(12);
7     *# Adjust learning rate $\lambda$ by gradient compensation*;
8     $\alpha\left(t\right) \leftarrow$ Computing by Eq.(14);
9     $\lambda\left(t\right) \leftarrow$ Computing by Eq.(15);
10    *# Optimize neural network by gradient-descent*;
11    $\theta \leftarrow \theta + \lambda\left(t\right)\frac{\partial J}{\partial \theta}$;
12 **end**

---

where $K_p$, $K_i$ and $K_d$ are the gain coefficients on the $P$, $I$ and $D$ terms, respectively. The coefficients $K_p$, $K_i$ and $K_d$ determine the contributions of present, past and future errors to the current correction. $e\left(t\right) = -RCE\left(t\right)$ represents the error of the $t$ th optimization epoch. Since $0 < \beta$, the updating step is described as follows:

$$\beta_t = \beta_{t-1}e^{-u(t)} \tag{12}$$

The PID controller is a widely employed feedback control mechanism in engineering and industrial processes [30, 28]. It is utilized in systems requiring precise control over variables. As depicted in Fig. 3, the PID controller is utilized in the model optimization to regulate RCE, ensuring balanced model calibration and mitigating overfitting compared to model accuracy within the baseline strategy. As depicted in (11), the PID comprises three terms: the proportional ($P$) term, integral ($I$) term, and derivative ($D$) term. Firstly, the proportional ($P$) action ensures that the controller responds proportionally to the current error. It provides an immediate correction to minimize the RCE. Secondly, the integral ($I$) action continuously integrates the error over time and adjusts the control signal accordingly, eliminating any steady-state error. It eliminates any steady-state error by gradually reducing the cumulative error to zero. Thirdly, the derivative ($D$) action anticipates the future behavior of the error by considering its rate of change, damping oscillations and overshoots to improve system stability and transient response. PID controllers play a crucial role in maintaining stability, accuracy, and efficiency in ensuring precise model calibration by adjusting probability-dependent gradient decay in model optimization.

### 3.2 Adaptive learning rate for gradient compensation

Changes in the probability-dependent gradient decay rate significantly impact model calibration because varying decay rates change dynamic characteristic on the optimization for different samples. While a large gradient decay rate can maintain a similar confidence level in optimizing both high-confidence and low-confidence samples, a small decay rate results in a curriculum learning sequence where confidence in low-confidence samples increases only when confidence in high-confidence samples reaches a certain threshold. Nonetheless, varying decay rates also lead to changing gradient magnitudes, thereby influencing model optimization. For instance, a small gradient at the outset of optimization could result in insufficient optimization, thereby diminishing the model's generalization. To mitigate the influence of gradient magnitude fluctuations on model optimization, we implement a dynamic learning rate to counteract the impact of gradient fluctuations caused by varying $\beta$.

Since $\left|\frac{\partial J}{\partial z_c}\right| + \sum_{j \neq c}\left|\frac{\partial J}{\partial z_j}\right| = 2\left|\frac{\partial J}{\partial z_c}\right|$, $\left|\frac{\partial J}{\partial z_c}\right|$ can represent the gradient magnitude of the sample in output layer. Building on (9), we establish a metric to quantify the magnitude of the gradient at that specific gradient decay rate $\beta$:

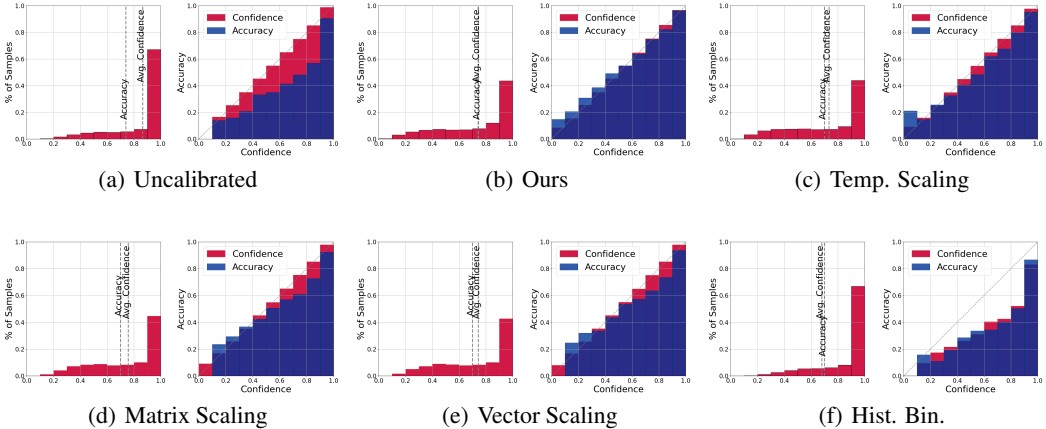

Figure 4: **Confidence histograms and reliability diagrams for different calibration methods with ResNet35 on CIFAR-100.** In each subplot, the left plot illustrates the sample distribution in individual bins, while the right plot displays the average confidence and accuracy in each bin. Our training calibration can improve performance on confidence estimate.

$$\alpha\left(t\right) = \int_0^1 \frac{1 - p_c}{1 + \left(\beta_t - 1\right)p_c}\,dp_c \tag{13}$$

Since $\beta > 0$, we obtain

$$\alpha\left(t\right) = \begin{cases} \frac{\beta_t \ln \beta_t - \beta_t + 1}{\left(\beta_t - 1\right)^2} & \beta_t \neq 1 \\ 0.5 & \beta_t = 1 \end{cases} \tag{14}$$

The learning rate, adjusted to account for the gradient dynamics over $t$ epoch, is modified according to the following formula:

$$\lambda\left(t\right) = \lambda\left(t-1\right)\alpha\left(t-1\right)/\alpha\left(t\right) \tag{15}$$

where $\lambda\left(t\right)$ is learning rate of $t$ th optimization epoch. The assumption underlying the gradient compensation proposed in (13)-(15) relies on the uniform distribution of samples within the probability interval $[0, 1]$. Integrating $\left|\frac{\partial J}{\partial z_c}\right|$ within the probability range $[0, 1]$ approximates the variation in gradient amplitudes across different gradient decay coefficients $\beta$. However, this assumption may not hold true during the optimization process. Nonetheless, this simple adaptive learning rate approach can effectively mitigate the issue of excessively small gradients resulting from overly large gradient decay rates. When the magnitude decreases, learning rate increases accordingly to compensate for the change in gradient amplitude. The whole Adaptive Probability-dependent Gradient Decay method by PID controller (PID-AGD) for model calibration is described in Algorithm 1.

## 4 Empirical experiments

The experimental validation comprises four main components. Firstly, we assess the performance of our algorithm and other calibration methods in terms of model calibration. Secondly, we compare the accuracy and model calibration results of our method with those obtained using different loss functions in supervised learning. Thirdly, we conduct ablation experiments on the proportional term $K_p$, integral term $K_i$, and differential term $K_d$ across PID controller. Finally, we conducted the ablation experiment with different optimizers to verify the effectiveness of the adaptive learning rate for gradient compensation.

**Train setting** The baseline models include ResNet and VGG variants. The datasets comprised SVHN, CIFAR-10/100, 102 Flower and Tiny-ImageNet. In CIFAR-10/100, the training set contained 40,000 images, with testing and validation sets comprising 10,000 images each. The ratio of training set, validation set and test set in 102 Flower is 2:1:1 respectively. For Tiny-ImageNet, 100,000 images were used for training, and 10,000 images for testing and validation. SVHN utilized 58,606 training

Table 1: The calibration performance of different post-hoc calibration methods. The best results are in bold, and relative improvements over $2^{nd}$ best result in each section are in red. Results are averaged over five runs with different seeds.

| Dataset | Model | Metric | Uncalibrated | Hist. Binning | Temp. Scaling | Vector Scaling | TS-AvUC | Ours |
|---|---|---|---|---|---|---|---|---|
| CIFAR-100 | ResNet18 | ECE | $0.160_{\pm 0.025}$ | $0.025_{\pm 0.006}$ | $0.033_{\pm 0.006}$ | $0.061_{\pm 0.012}$ | $0.028_{\pm 0.004}$ | $\mathbf{0.006}(\downarrow 0.019)$ |
| | | MCE | $0.344_{\pm 0.055}$ | $0.078_{\pm 0.012}$ | $0.059_{\pm 0.011}$ | $0.138_{\pm 0.022}$ | $\mathbf{0.052}(\downarrow 0.007)$ | $0.068\uparrow_{\pm 0.018}$ |
| | | AdaECE | $0.160_{\pm 0.023}$ | - | $0.030_{\pm 0.007}$ | $0.061_{\pm 0.011}$ | $0.027_{\pm 0.006}$ | $\mathbf{0.007}(\downarrow 0.020)$ |
| CIFAR-100 | ResNet35 | ECE | $0.172_{\pm 0.027}$ | $0.034_{\pm 0.009}$ | $0.026_{\pm 0.004}$ | $0.056_{\pm 0.011}$ | $0.035_{\pm 0.008}$ | $\mathbf{0.011}(\downarrow 0.015)$ |
| | | MCE | $0.351_{\pm 0.061}$ | $0.064_{\pm 0.010}$ | $\mathbf{0.053}(\downarrow 0.010)$ | $0.117_{\pm 0.019}$ | $0.146_{\pm 0.025}$ | $0.063\uparrow_{\pm 0.011}$ |
| | | AdaECE | $0.172_{\pm 0.028}$ | - | $0.027_{\pm 0.006}$ | $0.053_{\pm 0.010}$ | $0.034_{\pm 0.007}$ | $\mathbf{0.014}(\downarrow 0.013)$ |
| CIFAR-100 | ResNet50 | ECE | $0.186_{\pm 0.031}$ | $0.025_{\pm 0.004}$ | $0.030_{\pm 0.013}$ | $0.073_{\pm 0.021}$ | $0.052_{\pm 0.012}$ | $\mathbf{0.016}(\downarrow 0.009)$ |
| | | MCE | $0.407_{\pm 0.101}$ | $0.110_{\pm 0.015}$ | $0.091_{\pm 0.022}$ | $0.153_{\pm 0.036}$ | $0.116_{\pm 0.021}$ | $\mathbf{0.056}(\downarrow 0.035)$ |
| | | AdaECE | $0.186_{\pm 0.029}$ | - | $0.029_{\pm 0.012}$ | $0.071_{\pm 0.028}$ | $0.052_{\pm 0.010}$ | $\mathbf{0.006}(\downarrow 0.023)$ |
| CIFAR-100 | VGG16 | ECE | $0.240_{\pm 0.106}$ | $0.035_{\pm 0.002}$ | $0.029_{\pm 0.003}$ | $0.035_{\pm 0.006}$ | $0.044_{\pm 0.008}$ | $\mathbf{0.013}(\downarrow 0.016)$ |
| | | MCE | $0.508_{\pm 0.151}$ | $\mathbf{0.042}(\downarrow 0.002)$ | $0.093_{\pm 0.029}$ | $0.084_{\pm 0.009}$ | $0.101_{\pm 0.026}$ | $0.044\uparrow_{\pm 0.003}$ |
| | | AdaECE | $0.240_{\pm 0.106}$ | - | $0.029_{\pm 0.004}$ | $0.035_{\pm 0.006}$ | $0.044_{\pm 0.008}$ | $\mathbf{0.012}(\downarrow 0.017)$ |
| CIFAR-10 | ResNet35 | ECE | $0.054_{\pm 0.010}$ | $0.011_{\pm 0.001}$ | $0.015_{\pm 0.002}$ | $0.014_{\pm 0.003}$ | $0.015_{\pm 0.006}$ | $\mathbf{0.009}(\downarrow 0.002)$ |
| | | MCE | $0.300_{\pm 0.085}$ | $0.255_{\pm 0.102}$ | $0.121_{\pm 0.026}$ | $\mathbf{0.077}(\downarrow 0.030)$ | $0.121_{\pm 0.021}$ | $0.089\uparrow_{\pm 0.012}$ |
| | | AdaECE | $0.054_{\pm 0.011}$ | - | $0.014_{\pm 0.004}$ | $0.013_{\pm 0.002}$ | $0.013_{\pm 0.005}$ | $\mathbf{0.010}(\downarrow 0.003)$ |
| SVHN | ResNet18 | ECE | $0.021_{\pm 0.006}$ | $0.016_{\pm 0.002}$ | $0.009_{\pm 0.003}$ | $0.007_{\pm 0.002}$ | $0.010_{\pm 0.003}$ | $\mathbf{0.005}(\downarrow 0.002)$ |
| | | MCE | $0.286_{\pm 0.053}$ | $\mathbf{0.251}(\downarrow 0.013)$ | $0.313_{\pm 0.052}$ | $0.313_{\pm 0.069}$ | $0.315_{\pm 0.080}$ | $0.264\uparrow_{\pm 0.084}$ |
| | | AdaECE | $0.021_{\pm 0.006}$ | - | $0.010_{\pm 0.004}$ | $0.009_{\pm 0.002}$ | $0.013_{\pm 0.006}$ | $\mathbf{0.006}(\downarrow 0.003)$ |
| 102 Flower | ResNet50 | ECE | $0.101_{\pm 0.018}$ | $0.084_{\pm 0.012}$ | $0.086_{\pm 0.011}$ | $0.093_{\pm 0.015}$ | $0.075_{\pm 0.009}$ | $\mathbf{0.033}(\downarrow 0.042)$ |
| | | MCE | $0.231_{\pm 0.048}$ | $0.365_{\pm 0.066}$ | $0.180_{\pm 0.041}$ | $0.163_{\pm 0.043}$ | $0.165_{\pm 0.044}$ | $\mathbf{0.132}(\downarrow 0.031)$ |
| | | AdaECE | $0.100_{\pm 0.017}$ | - | $0.089_{\pm 0.012}$ | $0.098_{\pm 0.019}$ | $0.079_{\pm 0.011}$ | $\mathbf{0.031}(\downarrow 0.048)$ |
| Tiny-ImageNet | ResNet35 | ECE | $0.144_{\pm 0.022}$ | $0.033_{\pm 0.005}$ | $0.017_{\pm 0.003}$ | $0.053_{\pm 0.007}$ | $0.017_{\pm 0.003}$ | $\mathbf{0.009}(\downarrow 0.008)$ |
| | | MCE | $0.236_{\pm 0.052}$ | $0.055_{\pm 0.016}$ | $0.035_{\pm 0.014}$ | $0.093_{\pm 0.021}$ | $\mathbf{0.030}(\downarrow 0.005)$ | $0.035\uparrow_{\pm 0.006}$ |
| | | AdaECE | $0.143_{\pm 0.021}$ | - | $0.017_{\pm 0.004}$ | $0.054_{\pm 0.008}$ | $0.016_{\pm 0.002}$ | $\mathbf{0.010}(\downarrow 0.014)$ |
| VisDrone | YOLOv3 | ECE | $0.112_{\pm 0.011}$ | $0.079_{\pm 0.008}$ | $0.082_{\pm 0.007}$ | $0.091_{\pm 0.015}$ | $0.074_{\pm 0.010}$ | $\mathbf{0.044}(\downarrow 0.030)$ |
| | | MCE | $0.232_{\pm 0.033}$ | $0.325_{\pm 0.032}$ | $0.178_{\pm 0.025}$ | $0.148_{\pm 0.019}$ | $0.159_{\pm 0.023}$ | $\mathbf{0.136}(\downarrow 0.012)$ |
| | | AdaECE | $0.113_{\pm 0.012}$ | - | $0.085_{\pm 0.008}$ | $0.092_{\pm 0.017}$ | $0.078_{\pm 0.011}$ | $\mathbf{0.046}(\downarrow 0.032)$ |
| COCO | YOLOv3 | ECE | $0.115_{\pm 0.019}$ | $0.095_{\pm 0.011}$ | $0.091_{\pm 0.013}$ | $0.104_{\pm 0.016}$ | $0.092_{\pm 0.015}$ | $\mathbf{0.077}(\downarrow 0.014)$ |
| | | MCE | $0.224_{\pm 0.025}$ | $0.158_{\pm 0.025}$ | $0.158_{\pm 0.027}$ | $0.165_{\pm 0.028}$ | $\mathbf{0.164}(\downarrow 0.001)$ | $0.174_{\pm 0.030}$ |
| | | AdaECE | $0.117_{\pm 0.018}$ | - | $0.093_{\pm 0.014}$ | $0.105_{\pm 0.018}$ | $0.094_{\pm 0.018}$ | $\mathbf{0.078}(\downarrow 0.015)$ |

images, 14,651 validation images, and 26,032 testing images. VisDrone contains 5471 training images, 1548 validation images and 3190 testing images. COCO utilized 82,783 training, 40,504 validation, and 40,775 testing images. In all classification experiments, the learning rate, momentum and weight clipping were set to 0.1, 0.9 and Norm=3, respectively. The learning rate decreased to 10% at 40% and 80% of the iterations, with weight decay set to $10^{-4}$ and a total of 200 iterations. For Tiny-ImageNet, the learning rate was set to 0.01 with a batch size of 64. The number of bins in all calibration metric are set to 10. $P$, $I$ and $D$ in our method are set to 1, 0.1, 1.

## 4.1 Calibration performance with other calibration methods

The experiments in this subsection aim to validate the efficacy of our proposed calibration method against baseline calibration methods, which employs PID control of the gradient decay rate. We compare our approach with other post-processing calibration methods, including Histogram Binning [31], Temperature Scaling [10], Vector Scaling, and TS-AvUC [13]. The evaluation metrics employed include ECE [29], MCE, and Adaptive Expected Calibration Error (AdaECE) [12]. The datasets primarily consist of CIFAR-10/100, SVHN, 102 Flower, Tiny-ImageNet, VisDrone and COCO, while the models predominantly belong to the VGG, ResNet and YOLO series. The comprehensive experimental results are presented in Table 1. The results of the visualization of all methods in ResNet35 on data CIFAR-100 for confidence histograms and reliability diagrams are presented in Fig. 4.

The results presented in Table 1 demonstrate that all methods effectively enhance the calibration of the model. However, post-processing calibration methods rely on an optimized independent output-probability mapping, which doesn't alter the optimization process of the original model itself. Consequently, these methods can solely refine the probability distribution of the model output. Our proposed method surpasses other calibration techniques in terms of overall ECE, MCE, and AdaECE. Therefore, these experimental findings underscore the effectiveness of our approach in enhancing model calibration by dynamically adjusting the gradient decay rate during the model optimization.

Table 2: The calibration performance and accuracy of different objective functions. The best results are in bold. Results are averaged over five runs with different seeds.

| Methods | Models | CIFAR-10 | | | | CIFAR-100 | | | |
|---|---|---|---|---|---|---|---|---|---|
| | | ACC (%) | ECE | MCE | AdaECE | ACC (%) | ECE | MCE | AdaECE |
| Softmax | ResNet18 | $93.7_{\pm0.39}$ | $0.041_{\pm0.010}$ | $0.281_{\pm0.076}$ | $0.042_{\pm0.013}$ | $73.6_{\pm0.29}$ | $0.160_{\pm0.026}$ | $0.344_{\pm0.048}$ | $0.160_{\pm0.026}$ |
| | ResNet35 | $93.9_{\pm0.39}$ | $0.054_{\pm0.015}$ | $0.300_{\pm0.083}$ | $0.054_{\pm0.016}$ | $73.8_{\pm0.30}$ | $0.172_{\pm0.022}$ | $0.351_{\pm0.077}$ | $0.172_{\pm0.023}$ |
| | VGG16 | $92.1_{\pm0.41}$ | $0.066_{\pm0.022}$ | $0.339_{\pm0.091}$ | $0.068_{\pm0.023}$ | $69.2_{\pm0.26}$ | $0.233_{\pm0.054}$ | $0.476_{\pm0.112}$ | $0.236_{\pm0.053}$ |
| Cosface | ResNet18 | $93.9_{\pm0.45}$ | $0.053_{\pm0.011}$ | $0.352_{\pm0.072}$ | $0.055_{\pm0.013}$ | $74.2_{\pm0.51}$ | $0.185_{\pm0.046}$ | $0.501_{\pm0.162}$ | $0.183_{\pm0.050}$ |
| | ResNet35 | $\mathbf{95.6}_{\pm0.42}$ | $0.048_{\pm0.012}$ | $0.317_{\pm0.095}$ | $0.049_{\pm0.011}$ | $74.6_{\pm0.38}$ | $0.181_{\pm0.065}$ | $0.488_{\pm0.127}$ | $0.178_{\pm0.063}$ |
| | VGG16 | $92.7_{\pm0.58}$ | $0.067_{\pm0.019}$ | $0.390_{\pm0.101}$ | $0.068_{\pm0.020}$ | $71.4_{\pm0.52}$ | $0.238_{\pm0.081}$ | $0.567_{\pm0.125}$ | $0.233_{\pm0.085}$ |
| Center loss | ResNet18 | $94.5_{\pm0.41}$ | $0.038_{\pm0.009}$ | $0.337_{\pm0.075}$ | $0.040_{\pm0.008}$ | $74.1_{\pm0.30}$ | $0.082_{\pm0.013}$ | $0.222_{\pm0.071}$ | $0.085_{\pm0.015}$ |
| | ResNet35 | $95.5_{\pm0.51}$ | $0.043_{\pm0.010}$ | $0.280_{\pm0.099}$ | $0.045_{\pm0.012}$ | $74.2_{\pm0.31}$ | $0.098_{\pm0.031}$ | $0.250_{\pm0.096}$ | $0.101_{\pm0.030}$ |
| | VGG16 | $\mathbf{93.1}_{\pm0.41}$ | $0.034_{\pm0.009}$ | $0.349_{\pm0.083}$ | $0.034_{\pm0.010}$ | $\mathbf{72.1}_{\pm0.37}$ | $0.216_{\pm0.042}$ | $0.472_{\pm0.104}$ | $0.231_{\pm0.045}$ |
| DCA | ResNet18 | $91.9_{\pm0.32}$ | $0.020_{\pm0.006}$ | $0.156_{\pm0.038}$ | $0.022_{\pm0.007}$ | $72.1_{\pm0.25}$ | $0.047_{\pm0.011}$ | $0.156_{\pm0.024}$ | $0.049_{\pm0.012}$ |
| | ResNet35 | $92.3_{\pm0.43}$ | $0.035_{\pm0.012}$ | $0.186_{\pm0.046}$ | $0.034_{\pm0.010}$ | $73.1_{\pm0.28}$ | $0.067_{\pm0.021}$ | $0.184_{\pm0.051}$ | $0.066_{\pm0.023}$ |
| | VGG16 | $90.7_{\pm0.28}$ | $0.027_{\pm0.008}$ | $0.255_{\pm0.078}$ | $0.027_{\pm0.008}$ | $70.9_{\pm0.37}$ | $0.133_{\pm0.028}$ | $0.269_{\pm0.059}$ | $0.141_{\pm0.032}$ |
| Ours | ResNet18 | $\mathbf{95.0}_{\pm0.41}$ | $\mathbf{0.007}_{\pm0.002}$ | $\mathbf{0.078}_{\pm0.021}$ | $\mathbf{0.008}_{\pm0.001}$ | $\mathbf{74.3}_{\pm0.43}$ | $\mathbf{0.006}_{\pm0.002}$ | $\mathbf{0.068}_{\pm0.018}$ | $\mathbf{0.007}_{\pm0.002}$ |
| | ResNet35 | $95.6_{\pm0.51}$ | $\mathbf{0.009}_{\pm0.002}$ | $\mathbf{0.089}_{\pm0.012}$ | $\mathbf{0.010}_{\pm0.003}$ | $\mathbf{75.4}_{\pm0.39}$ | $\mathbf{0.011}_{\pm0.003}$ | $\mathbf{0.063}_{\pm0.011}$ | $\mathbf{0.014}_{\pm0.002}$ |
| | VGG16 | $92.6_{\pm0.35}$ | $\mathbf{0.011}_{\pm0.002}$ | $\mathbf{0.083}_{\pm0.031}$ | $\mathbf{0.012}_{\pm0.004}$ | $71.9_{\pm0.35}$ | $\mathbf{0.028}_{\pm0.008}$ | $\mathbf{0.044}_{\pm0.003}$ | $\mathbf{0.030}_{\pm0.010}$ |

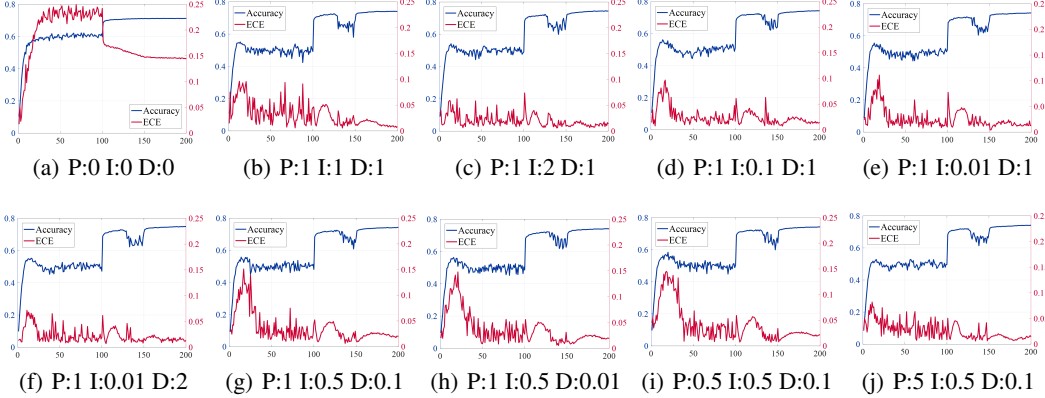

(a) P:0 I:0 D:0    (b) P:1 I:1 D:1    (c) P:1 I:2 D:1    (d) P:1 I:0.1 D:1    (e) P:1 I:0.01 D:1

(f) P:1 I:0.01 D:2    (g) P:1 I:0.5 D:0.1    (h) P:1 I:0.5 D:0.01    (i) P:0.5 I:0.5 D:0.1    (j) P:5 I:0.5 D:0.1

Figure 5: **Accuracy and ECE of different PID settings with ResNet35 on CIFAR-100.** The preceding figures illustrate the testing accuracy and ECE outcomes in the model optimization. Notably, the accuracy appears insensitive to the PID controller configuration. Nonetheless, excessive settings of $P$, $I$, and $D$ may compromise the stability of ECE during the model optimization.

## 4.2 Consistent optimization in supervised learning

Model accuracy and calibration are typically considered as two independent optimization metrics. In the baseline optimization strategy employing cross-entropy as the loss function, the model calibration tend to overfit earlier than the accuracy. To examine the impact of different optimization strategies on model calibration characteristics, Table 2 illustrates these effects. Comparative algorithms such as cosface [32], center loss [33], CE with DCA [34] and Softmax-based cross-entropy were utilized, and their performance in accuracy and uncertainty estimation was observed. While cosface, center loss, and our proposed approach, showed improvements in accuracy, the former two algorithms did not consider model calibration, resulting in overconfident predictions. DCA regularity improves model calibration but hurts accuracy. Conversely, our proposed PID-based method with variable gradient decay rate ensures both model accuracy and calibration. This reaffirms the significant influence of probability-dependent gradient decay rates on model calibration and overall performance.

## 4.3 Ablation experiments and analysis for PID controller

In the model optimization, both model optimization and gradient coefficient decay need consideration, constituting a bi-level optimization problem. From a control system perspective, altering the gradient decay rate modifies the dynamic characteristics of subsequent iterations in the model optimization. However, its impact on model calibration may not be fully apparent in the immediate optimization

Table 3: Different optimizer performance in ResNet35 on CIFAR-100. Results are averaged over five runs with different seeds. Adam optimization compromises model accuracy when applied to a dynamic optimization objective using a PID controller approach.

| SGD | Adam | PID Controller Approach | Gradient Compensation | Accuracy | ECE | AdaECE |
|-----|------|-------------------------|-----------------------|----------|------|--------|
| ✓ | - | - | - | 73.8% | 0.172 | 0.172 |
| ✓ | - | ✓ | - | 72.5% | 0.022 | 0.023 |
| - | ✓ | ✓ | - | 63.5% | 0.023 | 0.024 |
| ✓ | - | ✓ | ✓ | 74.7% | 0.012 | 0.013 |

results and may require more iterations for observation. Thus, viewed from this perspective, the entire control system can be regarded as having a time lag, whereby the controlled variable $\beta$ exhibits a certain delay concerning the RCE. While it may be challenging to mathematically describe the control system, the PID controller serves as a "black-box" controller, leveraging the integral and differential variations of the error, proving highly effective. The ablation experiments are shown in Fig. 5. We conclude that model calibration is robust to the choice of PID parameters; however, setting the PID parameters too high can compromise the stability of ECE during model optimization. Based on this, we selected PID settings of 1, 0.1, and 1 for the experiments presented above. On the other hard, in Fig. 5, the accuracy curves with our method all exhibited a noticeable jitter, which requires further investigation.

### 4.4 Ablation experiments of different optimizer

The motivation for proposing the adaptive learning rate for gradient compensation arises from the observation that significant variations in gradient magnitude can negatively impact the model's optimization for classification error. A dynamic learning rate helps maintain a relatively stable gradient magnitude. Additionally, the Adam optimizer aids in reducing gradient fluctuations. To verify the novelty of our method, Table 3 presents the performance of various optimizers when applied to our proposed PID controller-based calibration method.

Our experimental results indicate that Adam can indeed provide a more stable gradient and calibration performance, particularly in conjunction with our PID controller approach for model calibration. However, it is notable that Adam results in reduced accuracy, achieving only 63.5% on CIFAR-100 with ResNet35, significantly lower than the baseline accuracy of 73.8%. A key difference arises in the baseline case handled by Adam. In our proposed PID controller method, which adjusts the hyperparameter $\beta$ during model calibration, the loss function is dynamic. While Adam retains previous gradient information, this can conflict with the current gradient vector direction because the optimization objective is dynamic. In contrast, our compensation method only modifies the learning rate and retains gradient vector direction pertinent to the current loss function. This may explain why the Adam optimizer does not yield better results.

## 5 Conclusion

The gradient decay rate plays a crucial role in shaping the calibration characteristics and uncertainty distribution of deep learning throughout the dynamic optimization. Our results show a negative correlation between the gradient decay rate with increasing instance-level probability and the overall confidence distribution. This paper introduces a novel optimization approach aimed at regulating the gradient decay rate hyperparameter $\beta$, via a PID controller. The goal is to achieve perfect model calibration by monitoring the proposed relative calibration error of the validation set. Within this control system framework as shown in Fig. 3, the probabilistic gradient decay rate serves as the controlled variable, while a newly defined relative calibration error acts as the control system output, mitigating both over-confidence and under-confidence in the model. Additionally, to address fluctuations of gradient amplitude resulting from varying gradient decay rate, a new learning rate compensation mechanism is employed. Empirical validation demonstrates that our proposed adaptive gradient decay rate optimization strategy, facilitated by a PID controller, effectively enhances both the accuracy and model calibration in deep learning, ensuring adequate calibration throughout the supervised learning.

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

# A Appendix

## A.1 Experiments compute resources

For experiments, we utilized compute resources featuring an NVIDIA A100 GPU with PCIe interface and 40GB memory capacity, accompanied by PyTorch version 1.7.0 with CUDA version 11.0. The computational backbone was supported by an Intel Xeon Gold 6278C processor.

## A.2 Experiments for fixed gradient decay rate

More empirical results validating the relationship between gradient decay hyperparameter $\beta$ and model calibration are presented in Tab. 4, Tab. 5 and Tab. 6, while confidence histograms and reliability diagrams are depicted in Figs. 6-9. These experiments consistently conclude that the gradient decay rate $\beta$ is negatively correlated with the overall confidence level of the model. Specifically, when the value $\beta$ is small, the gradient decay rate decreases, leading to higher overall confidence in the model output and a greater likelihood of overconfident probabilistic output. Conversely, as the value $\beta$ increases, the gradient decay rate increases, resulting in lower overall confidence in the model's output and a higher probability of underfitting in the model's probabilistic output distribution. Although model accuracy appears somewhat linked to this hyperparameter, conclusive generalizations from theoretical perspective cannot be drawn for the current experimental results.

Table 4: Model calibration of different gradient decay and post-processing calibration. The best results are in bold. Results are averaged over five runs with different seeds. ($bins = 10$)

| Dataset | Model | Metric | Gradient decay factor $\beta$ | | | | Vector Scaling | Temp. Scaling |
| --- | --- | --- | --- | --- | --- | --- | --- | --- |
| | | | 20 | 10 | 1 | 0.1 | | |
| CIFAR-100 | ResNet18 | ECE | **0.019**±0.003 | 0.048±0.008 | 0.111±0.011 | 0.161±0.021 | 0.039±0.006 | 0.026±0.005 |
| | | MCE | **0.063**±0.011 | 0.139±0.025 | 0.306±0.051 | 0.423±0.076 | 0.135±0.036 | 0.064±0.021 |
| CIFAR-100 | ResNet34 | ECE | **0.026**±0.004 | 0.055±0.008 | 0.131±0.019 | 0.182±0.022 | 0.042±0.006 | 0.038±0.005 |
| | | MCE | 0.087±0.011 | 0.162±0.031 | 0.233±0.068 | 0.332±0.091 | 0.131±0.032 | **0.059**±0.011 |
| CIFAR-100 | VGG16 | ECE | 0.122±0.009 | 0.163±0.011 | 0.207±0.031 | 0.226±0.033 | 0.030±0.008 | **0.022**±0.005 |
| | | MCE | 0.317±0.021 | 0.378±0.051 | 0.499±0.088 | 0.556±0.093 | 0.523±0.109 | **0.041**±0.011 |
| CIFAR-10 | ResNet18 | ECE | 0.021±0.004 | 0.025±0.006 | 0.036±0.010 | 0.042±0.011 | **0.011**±0.003 | 0.015±0.003 |
| | | MCE | 0.591±0.153 | 0.268±0.095 | 0.295±0.068 | 0.355±0.111 | **0.051**±0.012 | 0.089±0.009 |
| Tiny-ImageNet | ResNet34 | ECE | **0.014**±0.002 | 0.036±0.005 | 0.089±0.011 | 0.226±0.056 | 0.017±0.006 | 0.019±0.004 |
| | | MCE | **0.035**±0.005 | 0.069±0.009 | 0.166±0.021 | 0.382±0.079 | 0.036±0.010 | 0.063±0.013 |
| Tiny-ImageNet | ResNet50 | ECE | 0.041±0.007 | **0.013**±0.002 | 0.104±0.021 | 0.188±0.032 | 0.023±0.004 | 0.027±0.003 |
| | | MCE | 0.082±0.011 | 0.044±0.009 | 0.149±0.020 | 0.377±0.045 | **0.039**±0.011 | 0.067±0.016 |

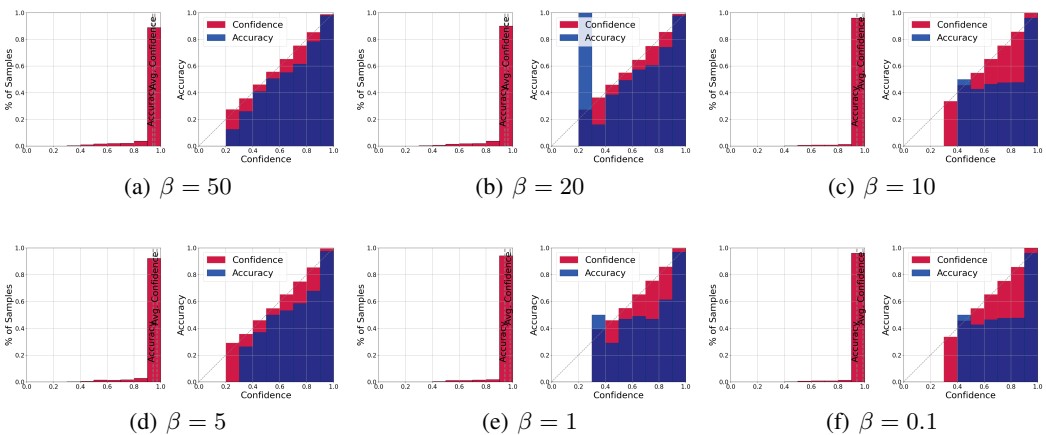

Figure 6: Confidence histograms and reliability diagrams for gradient decay with ResNet18 on CIFAR-10. ($bins = 10$)

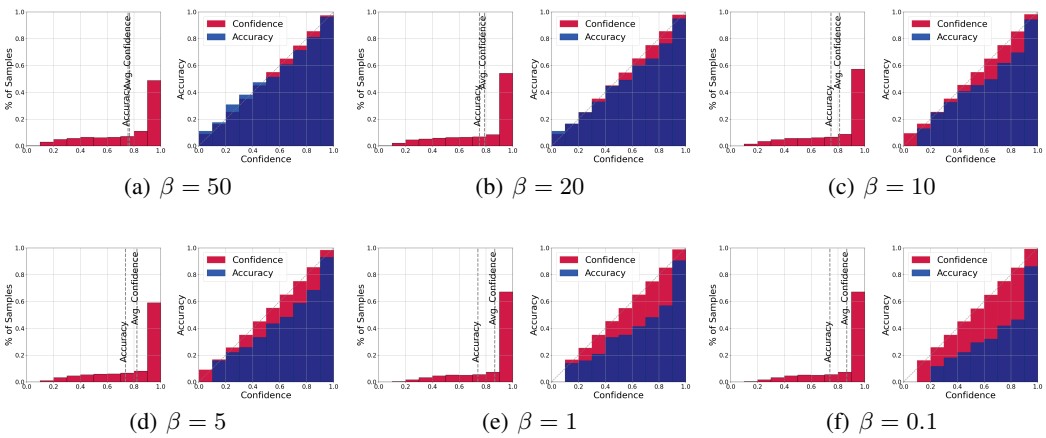

Figure 7: Confidence histograms and reliability diagrams with ResNet34 on CIFAR-100. ($bins = 10$)

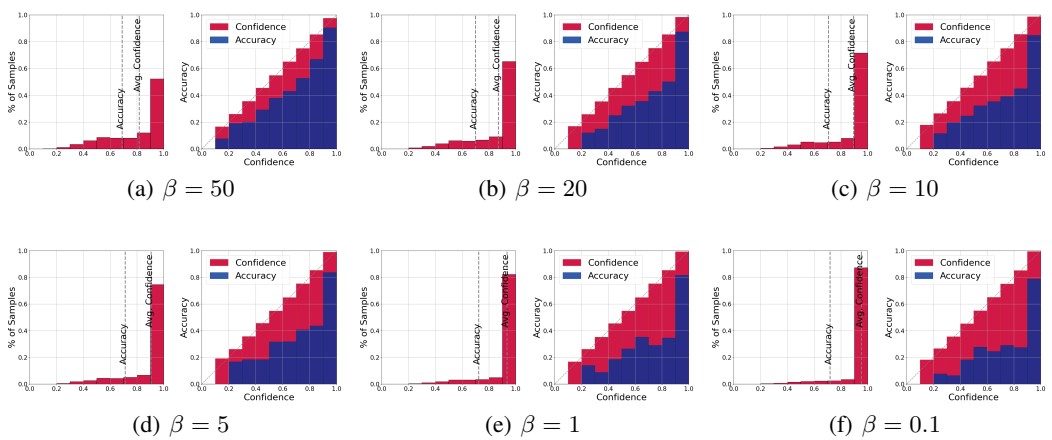

Figure 8: Confidence histograms and reliability diagrams for gradient decay with VGG16 on CIFAR-100. ($bins = 10$)

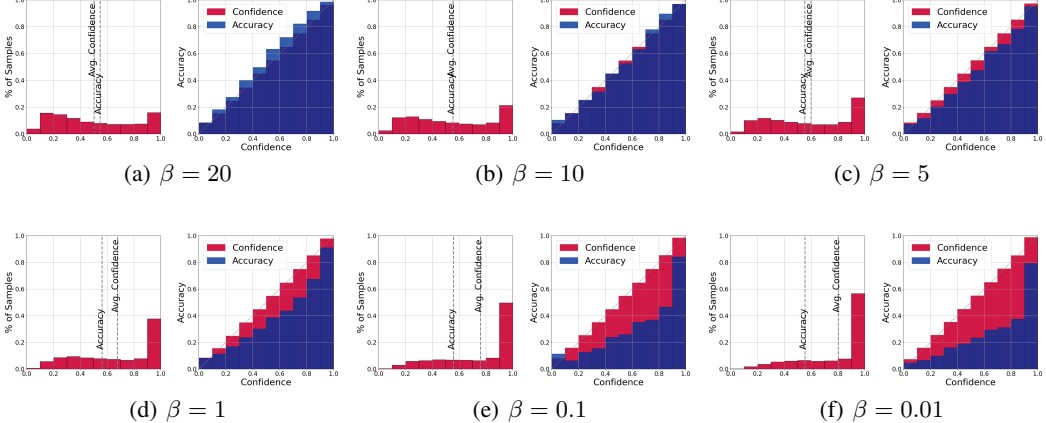

Figure 9: Confidence histograms and reliability diagrams with ResNet50 on Tiny-ImageNet. ($bins = 10$)

Table 5: The performance of ResNet34 on Tiny-ImageNet with different gradient decay. The best results are in bold. Results are averaged over five runs with different seeds. ($bins = 10$)

| Metric | Gradient decay factor $\beta$ | | | | | |
|---|---|---|---|---|---|---|
| | 20 | 10 | 5 | 1 | 0.1 | 0.01 |
| Top-1 Acc (%) | 52.8 | 53.2 | **53.8** | 53.4 | 53.7 | 53.6 |
| Top-5 Acc (%) | **75.8** | 75.5 | 75.1 | 75.0 | 74.4 | 73.2 |
| Training Acc (%) | 88.6 | 88.5 | 89.7 | 90.3 | **90.9** | 90.5 |
| ECE ($bins = 10$) | **0.015** | 0.034 | 0.076 | 0.087 | 0.224 | 0.274 |
| MCE ($bins = 10$) | **0.036** | 0.065 | 0.151 | 0.176 | 0.406 | 0.518 |

Table 6: The performance of ResNet50 on Tiny-ImageNet with different gradient decay. The best results are in bold. Results are averaged over five runs with different seeds. ($bins = 10$)

| Metric | Gradient decay factor $\beta$ | | | | | |
|---|---|---|---|---|---|---|
| | 20 | 10 | 5 | 1 | 0.1 | 0.01 |
| Top-1 Acc (%) | 55.9 | 56.0 | **56.4** | 56.3 | 56.4 | 56.2 |
| Top-5 Acc (%) | 77.7 | **78.0** | 77.3 | 76.6 | 76.0 | 74.9 |
| Training Acc (%) | 86.4 | 88.8 | 90.3 | 91.9 | **92.3** | 91.8 |
| ECE ($bins = 10$) | 0.045 | **0.014** | 0.046 | 0.114 | 0.203 | 0.249 |
| MCE ($bins = 10$) | 0.084 | **0.044** | 0.082 | 0.151 | 0.388 | 0.476 |

## A.3 Limitations and future works

Our work currently lacks theoretical analysis. Although all experimental findings consistently demonstrate the impact of gradient decay rate $\beta$ on model calibration, we still require theoretical frameworks to explain how the gradient decay rate affects the overall confidence distribution. Our experiments indicate that large gradient decay rates result in similar confidence levels across samples, while smaller rates yield more discriminatory levels. In essence, smaller decay rates enforce a more stringent curriculum learning sequence, whereby increased confidence in difficult samples only occurs after optimizing easier ones. Consequently, this leads to greater differentiation in final confidence distribution of different samples. While these empirical observations are compelling, they lack theoretical substantiation.

Methodologically, our current approach employs a PID controller to control gradient decay in optimization. However, in practice, the effect of gradient decay rate adjustments on model calibration requires a large number of epochs to manifest changes. Viewed from a control systems perspective, this delay indicates a substantial time lag in the control object. However, as neural network optimization processes defy mathematical description, designing effective controllers becomes inherently challenging. Furthermore, our work also lacks comprehensive statistical analysis of the calibrated outputs. Future research should address how alterations in the dynamic gradient decay rate impact the internal optimization process.

Additionally, the temperature coefficient $\tau$ also impacts model calibration. The Softmax with small $\tau$ disperses the inter-class distance by adjusting the probability output to focus more on hard negative samples. Nevertheless, large $\tau$ can only smooth the output of all categories and cannot mine more information from simple positive samples. On the contrary, small $\beta$ makes the gradient decay slowly so that easy positive samples can be sufficiently learned up to high confidence. An appropriate $\beta$ can mining more discriminative features on the whole. Similarly, large $\beta$ only keeps the samples at the same level of confidence and cannot extract more meaningful features from challenging samples. $\tau$ and $\beta$ improved the mining capability of Softmax in two different dimensions. The exploration of optimizing the effects of both hyperparameters to improve model calibration represents a promising approach.

