# OpenReview forum: "A PID Controller Approach for Adaptive Probability-dependent Gradient Decay in Model Calibration"
_NeurIPS.cc/2024/Conference — NeurIPS 2024 poster_

### Official Review · Reviewer_3YTY · 2024-07-04

**Soundness:** 2
**Presentation:** 3
**Contribution:** 2
**Rating:** 6
**Confidence:** 3

**Summary:**

The submitted paper proposes a PID-based controller to ensure the consistent optimisation of model accuracy and model calibration. The controller with proposed Relative Calibration Error (RCE) dynamically adjust gradient decay rate to "control" model confidence. By applying a learning rate compensation mechanism, the side effects of the dynamic gradient decay rate, such as fluctuations in gradient amplitude, can be mitigated.

**Strengths:**

1, The proposed Relative Calibration Error (RCE) is a simple but highly efficient way to exhibit over-confidence and under-confidence, which can be exploited by PID-based controller. \
2, A very clear illustration (Figure 3) is used to connect PID controller and model optimisation.\
3, Empirical experiments show good results for the proposed method.\
4, The potential application (dynamic calibration) on online machine learning.

**Weaknesses:**

1, This work lacks theoretical analysis as authors mentioned in the appendix.\
2, Please refer to the "Questions" part.

**Questions:**

1, Have you thoroughly investigated any related research about PID on model calibration since no related work section in this paper? PID controller is a classical and historical method. There are several researches related to PID and model training but I'm not sure whether there are similar researches of this work. Additionally, in "$\textit{Line 68-69}$ [29]", I guess you may want to highlight the success of PID controller in the machine leaning filed (Interdisciplinary filed) to enhance the feasibility of this work.\
2,  $\textit{Line 102}$ : The temperature coefficient $\tau$ is introduced as an hyper-parameter. However, it seems this hyper-parameter is fixed to 1 ($\textit{Line 115}$) without any further discussion. Have you have tried any different value or any methods to optimise this hyper-parameter?\
3, Have you tried any different optimisers especially any adaptive optimisers like Adam since you want to mitigate the gradient fluctuation ,or only SGD with the proposed adaptive learning rate strategy is applied as you mentioned in $\textit{Algorithm 1}$ (Page 7)?\
4, What is the strategy for selecting/tuning the value of P/I/D terms?\
5, Could you please explain more details about the statement (threshold related) in $\textit{Line 178-179}$\
6, How did you fine-tune the comparative methods where you mentioned in $Table$ 1? Have you chosen the best result of other methods in their experiments to compare yours?\
$\textbf{(Possible) Minor issues:}$\
1, $\textit{Abstract}$ : you mentioned "During model optimization, the expected calibration error tends to overfit earlier than classification accuracy, indicating distinct optimization objectives for classification error and calibration error" which points the accuracy and calibration are two distinct but related definitions. Maybe you can briefly describe the difference between them in Introduction section for more general audience.\
2, $\textit{Line 50}$ : If I understand correctly, "These methods" represents the "Training-based model calibration methods". If it is, maybe you can reaffirm the subject instead of pronoun since you use singular from in previous sentences (Line 48 and 49).\
3, $\textit{Line 96}$ : "$D_b$ contains all sample with $\hat{p} \in [\frac{b}{M},\frac{b+1}{M})$" but you also mentioned the $b$ starts from $1$ to $M$. It seems the first bin $[0,\frac{1}{M}]$ and the right boundary of the last bin $[..., 1]$ are missed. One simple way is to change it to $\hat{p} \in [\frac{b-1}{M},\frac{b}{M})$ and additionally make the last bin compact if I understand it correctly.\
4, $\textit{Line 98}$ : "ECE and MCE". Better to use the full term when you first mention it in the main text. For example, "Expected Calibration Error (ECE) and Maximum Calibration Error (MCE)".\
5, Changing the label of y-axis "\% of samples" in $Figure$ 2 to "$\times100$% of samples" may more precise?\
6, The label of x-axis in $Figure$ 5 is missed. Is that "Epoch"?\

**Limitations:**

Please see the weaknesses part.

---

> ### Author Rebuttal · Authors · 2024-08-04
>
> **Answers to Questions**
> * To the best of my knowledge, no existing work addresses model calibration using PID control. Most prior approaches apply PID concepts to optimization problems rather than model calibration. Our work, however, establishes a connection between model calibration and gradient decay rate by introducing a probability-dependent gradient decay coefficient. We then implement a model calibration strategy using the PID method by controlling this gradient decay rate. This novel approach introduces a probabilistically related gradient decay rate, which is controlled to achieve effective model calibration. To the best of our knowledge, this is the first discussion of the probability-dependent gradient decay rate in relation to model calibration.
>
> * The temperature coefficient and gradient decay rate have distinct effects on model optimization. As the temperature coefficient increases, the variability in confidence levels among different samples decreases, causing their confidence levels to converge. Conversely, a smaller temperature coefficient results in greater variation in confidence levels among samples. Temperature has been intensively studied in areas such as model calibration and knowledge distillation aspects, and there are many well-established optimization methods. Our method does not address the optimization of the temperature coefficient. We appreciate the reviewer for highlighting this important issue. We will jointly consider the effects of temperature and probability-dependent on model calibration in future work.
>
> * We thank the reviewer for the insightful question. We have tested the Adam optimizer to assess its ability to provide a relatively stable gradient for model parameter optimization. Below is a demo experiment. Our experiments were conducted on CIFAR-10 and CIFAR-100 using ResNet and VGG networks. The results indicate that, while Adam offers a stable gradient, its accuracy is lower compared to the SGD optimizer with dynamic gradient decay coefficients and our proposed gradient compensation approach. For example, Adam achieved only 63.5% accuracy on CIFAR-100 with ResNet35, which is significantly lower than the baseline accuracy of 73.8%.
>
> | SGD       | Adam      | PID Controller Approach | Gradient Compensation | Accuracy | ECE   | AdaECE |
> | --------- | --------- | ----------------------- | --------------------- | --------| ----- | ------ |
> | ✓ | - | -  | -| 73.8%   | 0.172 | 0.172  |
> | ✓ |- | ✓| - | 72.5%   | 0.022 | 0.023  |
> | - | ✓| ✓| - | 63.5%   | 0.023 | 0.024  |
> | ✓  | - | ✓| ✓| 74.7%   | 0.012 | 0.013  |
>
> A key difference arises in the baseline case handled by Adam. In our proposed PID controller method, which adjusts the hyperparameter $\beta$ during model calibration, the loss function is dynamic. While Adam retains previous gradient information, this can conflict with the current optimization direction. In contrast, our compensation method only modifies the learning rate and retains gradient information pertinent to the current loss function. This may explain why the Adam optimizer does not yield better results.
>
> * The current PID settings are based on a trial-and-error approach. However, it is important to note that varying the PID setting does not significantly affect model accuracy or calibration, as demonstrated by the ablation experiments with different P/I/D settings shown in Figure 5. Different PID settings provide effective calibration results and maintain model accuracy. In summary, our method is robust to variations in P/I/D settings.
>
> * The hyperparameter $\beta$ is designed to correlate with the overall confidence level of the samples, as illustrated in Figure 2 and Figures 6-9 in the Appendix, as well as Tables 4-6. Additionally, Equation (5) shows that the cross-entropy loss function in conjunction with Softmax can be approximated using a max function, where $\beta$ serves as a threshold. This is further illustrated by Equation (5).
>
> **Confidence Distribution of Samples with Different Gradient Decay of Three-Layer FCNN on MNIST**
>  The `#` indicates the number of samples that belong to the confidence interval.
>
> | Gradient Decay Factor |1|0.5|0.1|0.01|0.001|
> |-----------------------|-----|-----|-----|------|-------|
> | #${p_c}\le 0.2$|903|828|1105|1325|2375|
> | #$0.2<{p_c}\le0.4$|454|206|119|91|142|
> | #$0.4<{p_c}\le0.6$|528|245|132|92|116 |
> | #$0.6<{p_c}\le0.8$|1291|484|191|100|193|
> | #$0.8<{p_c}\le1$|56824|58237|58453|58362|57147|
>
> We will also give an example to illustrate this point. We give the confidence distribution of the MINST dataset in a fully connected network for the training set samples under varying $\beta$. Our experiments reveal that a smaller $\beta$ results in a higher overall confidence level in the distribution, such $\beta=0.5$. However, when $\beta$ exceeds a certain threshold $\beta>=0.1$, the number of high-confidence samples decreases. This phenomenon occurs because a small decay rate creates a curriculum learning sequence, where the confidence in low-confidence samples only increases once the confidence in high-confidence samples surpasses a soft threshold. If this threshold is too high, the confidence of high-confidence samples continues to increase, while low-confidence samples fail to achieve a higher confidence level.
>
> We also give another chart in the added PDF response, which can also give more details about the statement.
>
> * We fine-tuned the hyperparameters for all the compared methods, including the learning rate for post-processing techniques, and selected the best results from these experiments.
>
> * We thank the reviewer for the careful review and for these minor questions and suggestions! We will address all minor issues raised and revise the presentation accordingly.

---

> > ### Comment · Reviewer_3YTY · 2024-08-12
> >
> > Thank you for your detailed rebuttal, which addresses my concern. I have decided to raise my score.

---

> > > ### Author Response · Authors · 2024-08-13
> > > **Thank you**
> > >
> > > Thank you for your time and valuable feedback. We are glad that our responses addressed your questions.

---

### Official Review · Reviewer_dGQ5 · 2024-07-06

**Soundness:** 2
**Presentation:** 3
**Contribution:** 2
**Rating:** 6
**Confidence:** 5

**Summary:**

The paper presents an approach to ensure consistent optimisation of both model accuracy and calibration. The authors used a PID-based controller for the task. The PID-based controller adjusts the gradient decay rate, which ultimately optimises the neural network by gradient descent. Further ablation studies have shown that the PID controller was effective in controlling the accuracy and ECE of the model.

**Strengths:**

1. The paper is written in a very good format, covering every aspect.
2. Proper equations and graphs are provided for a better understanding of the paper.
3. The method is compared against other post-processing calibration methods, and it shows better results than them.
4. The ablation study further provides deeper understanding of how the variation of the parameters of the controller effects the performance.

**Weaknesses:**

1. The combination of PID controllers with gradient decay for model calibration appears to be an incremental improvement rather than a groundbreaking innovation. The paper does not sufficiently differentiate its contribution from existing methods using PID controllers in optimization tasks.
2. The paper claims to address both over-confidence and under-confidence in model predictions. However, the analysis of how well the proposed method balances these two aspects is not thoroughly presented. The paper should include more detailed experiments and discussions on this balance.

**Questions:**

1. How sensitive is the proposed method to the choice of PID controller parameters? Could you provide a sensitivity analysis or guidelines for selecting these parameters?
2. Why were CIFAR-10/100 and Tiny-ImageNet chosen for the experiments? Have you considered testing on other datasets, especially those from different domains, to demonstrate the generalizability of your method?

**Limitations:**

1. The study does not address the scalability of the proposed method to larger and more complex datasets or models. The computational overhead introduced by the PID controller and adaptive learning rate adjustments is not discussed.
2. The paper suggests that the proposed method can prevent overconfidence, but it does not adequately address how it handles overfitting to the training data. A more detailed analysis of overfitting prevention mechanisms is needed.

---

> ### Author Rebuttal · Authors · 2024-08-04
>
> **Answers to Questions**
>
> * In the experiments detailed in Sections 4.1 and 4.2, the hyperparameters of the PID controllers were determined through trial-and-error. In Section 4.3, we present ablation experiments that explore various P/I/D hyperparameters in the PID controller. As shown in Figure 5, the results from these ablation experiments demonstrate that the calibration results and model accuracy are insensitive to hyperparameter settings. Different hyperparameter settings in the PID controller yield better model calibration results and produce different outcomes compared to models without PID controller. In summary, our method is robust to variations in P/I/D settings as shown in Section 4.3.
>
> * In the original submission, we validated our approach on image classification tasks using CIFAR-10, CIFAR-100, SVHN, 102 Flowers, and the Tiny-ImageNet. The above datasets have been widely used in experiments and their results are representative. As suggested by the reviewer, we applied our proposed PID Controller Approach to the object detection tasks in VisDrone and COCO, where YOLOv3 serves as the object detection model. The confidence of the object classification was calibrated, and the experimental results validated the effectiveness of our approach.
>
> |Dataset|Model|Metric|Uncalibrated|Hist. Bin.|Temp. Scaling|TS-AvUC|Ours|
> |:----------|:------:|:------:|-------------:|-----------:|--------------:|--------:|------:|
> |VisDrone|YOLOv3|ECE|0.101|0.084|0.086|0.075|0.043|
> |||MCE|0.231 | 0.365 |0.180|0.165|0.142|
> |||AdaECE |0.100|-|0.089|0.079|0.046|
> |COCO|YOLOv3|ECE|0.121|0.101|0.093|0.091|0.081|
> |||MCE|0.236|0.169|0.165|0.154|0.184|
> |||AdaECE|0.126|-|0.096|0.092|0.082|
>
> **Rebuttal to Weakness**
>
> **_The motivation and novelty of our work:_** Consider the reviewer's comment in the "Weakness" section. We would like to provide further explanation about the motivation and novelty of our approach for the reviewer and clarify how it differs from other PID-based optimization study.
>
> Most previous PID methods are applied to model optimization, such as gradient optimization. Moreover, many optimization techniques, like momentum and Adam, share similarities with PID control concepts. However, the problem we aim to address is not only related to optimization or expediting hyperparameter tuning but rather to ensuring model calibration during the optimization. To the best of our knowledge, no existing work has employed a PID controller approach for model calibration, making our work distinct from prior PID-based optimization strategies. We are tackling a different problem by focusing on model calibration.
>
> In addition to utilizing a PID control method, the primary novelty of our work is the introduction of a probability-dependent gradient decay coefficient. In this paper, we have verified the relationship between this coefficient and model calibration both deductively and empirically, as demonstrated in Figures 1-2, Equations (4-9), and the experiments detailed in the Appendix. This approach provides a new perspective on model calibration, shedding light on the calibration issues prevalent in modern neural networks.
>
> Overall, this work introduces a novel approach to calibration that significantly differs from previous methods. As discussed in the paper, the connection between the gradient decay rate and model calibration offers a new explanation for the overconfidence of modern models. To the best of our knowledge, this is the first discussion of the probability-dependent gradient decay rate in relation to model calibration.
>
> **_How is calibration implemented using a PID controller approach?_**  We introduce a probability-dependent gradient decay coefficient into the loss function. This coefficient regulates the rate at which the gradient of a sample decays as the confidence level increases, as illustrated in Figure 1 and Equations (4-9). Figures 2, 6-9 in the Appendix, and Tables 4-6 demonstrate that the rate of gradient decay negatively correlates with the confidence distribution of sample pairs passing through the model. The experiments described above clarify the different phenomena of overconfidence and underconfidence exhibited by model confidence when various rates of gradient decay are chosen.  Please refer to Figure 2 and Figures 6-9. This observation motivated the use of a PID control method to manage the confidence levels of samples.
>
> During model optimization, we monitor the average confidence level of the dataset in real-time, thereby ensuring confidence calibration through PID control. Our work calibrates model by managing a single probability-dependent gradient decay rate and presents an innovative approach to this problem.
>
> **Rebuttal to Limitation**
> * **_Computational overhead of our approach:_** In our approach, there is just a single hyperparameter that needs to be tuned. This adjustment comes from the computation of the relative calibration error with respect to the validation set. Therefore, compared to the baseline optimization strategy, the additional computation is very small, which is negligible for the whole optimization process.
>
> * **_The scalability of the proposed method:_** In our original experiments, we evaluated our method on CIFAR-10, CIFAR-100, SVHN, FLOWER102, and Tiny-ImageNet datasets across four models and nine different methods. The publicly available datasets described above are representative. In response to the reviewer's suggestions and concerns, we applied our method to object detection tasks and demonstrated its effectiveness from different domains. See "Answers to Questions."
>
> * **_Overfitting to the training data:_** Our paper focuses on the model calibration problem. The problem of overfitting for the training set is not in our consideration, but we will consider the impact of our proposed method of dynamically adjusting the gradient decay rate on overfitting in future work. Thanks to the reviewer for the suggestion.

---

> > ### Comment · Reviewer_dGQ5 · 2024-08-11
> >
> > Dear Authors,
> >
> > Thank you for your detailed and thoughtful responses to my comments and questions. I appreciate the additional experiments and explanations you provided to address my concerns. Below, I offer some final thoughts on your rebuttal:
> >
> > Sensitivity to PID Controller Parameters:
> >
> > I appreciate the thorough explanation and the ablation studies provided in Section 4.3. It's reassuring to know that the method demonstrates robustness to variations in the PID controller parameters. Including these details in the final paper will be valuable for readers who may want to implement your approach in various contexts.
> >
> >
> >
> > Choice of Datasets:
> >
> > Thank you for conducting additional experiments on VisDrone and COCO datasets. These results further strengthen your paper by demonstrating the generalizability of your approach across different domains. Including these results in the final submission will undoubtedly enhance the paper's impact.
> >
> >
> > Motivation and Novelty:
> >
> > Your clarification on the novelty of your work, especially the focus on model calibration rather than mere optimization, is well received. The distinction you draw between your approach and previous PID-based methods is clear, and I believe this will help in positioning your work as a significant contribution to the field. Highlighting these points more prominently in the paper, particularly in the introduction and conclusion, will help readers better understand the uniqueness of your contribution.
> >
> >
> > Computational Overhead and Scalability:
> >
> > Your explanation that the computational overhead introduced by your method is minimal and that it scales well with larger datasets addresses my concern effectively. Including a brief discussion on this in the final paper, particularly in the limitations or methodology section, would be beneficial for readers concerned with the practical implementation of your approach.
> >
> >
> > Overfitting Concerns:
> >
> > I understand that your focus was primarily on model calibration, and I appreciate your acknowledgment of the potential impact on overfitting. I encourage you to explore this aspect in future work, as understanding how your method interacts with overfitting dynamics could further solidify its utility in practical applications.
> >
> >
> > In conclusion, I believe your paper has made significant strides in addressing my initial concerns, and the additional work you've done to clarify and expand on your methodology is commendable. I look forward to seeing these updates reflected in the final version of your paper and I am increasing your score from my side

---

> > ### Author Response · Authors · 2024-08-12
> > **Thank you**
> >
> > Thank you for your time and valuable feedback. We are glad that our responses addressed your questions.

---

### Official Review · Reviewer_iW8b · 2024-07-16

**Soundness:** 3
**Presentation:** 2
**Contribution:** 3
**Rating:** 6
**Confidence:** 3

**Summary:**

The authors propose a method for improving the calibration of neural networks, which are known to be overconfident in their predicitions. Their method is based on modifying the softmax function to include a tunable hyperparameter- which they call the gradient decay coefficient- which is controlled throughout the optimization by assessing the level of calibration of the model on a validation set and tuning the gradient decay coefficient using a PID controller. They also propose an adaptive learning rate scheduler to ensure that the changing of the gradient decay coefficient doesn't result in vanishing gradients. The authors conduct empirical experiments to validate the effectiveness of their PID controller based approach compared to other calibration methods.

**Strengths:**

The issue of calibration and the mitigation of overconfident predictions by neural networks is clearly important, and the authors have proposed what strikes me as a sensible method to tackle it. PID controllers are widely used in industry, but remain underutilized in ML applications, so their use by the authors is refreshing. The empirical experiments conducted by the authors seem adequate to demonstrate their claims.

**Weaknesses:**

The definition of the gradient decay coefficient is not well motivated- it is not clear to me why this particular modification of the softmax is to be preferred over others. It is likewise not clear to me what motivates the specific form of the learning rate scheduler in formula (15). The authors ackowledge that their method does not currently have theoretical justification, which is understandable given the general lack of solid theory for deep learning, but given this situation it is hard to assess whether the proposed method is likely to succeed in more general settings.

**Questions:**

Did the authors try different optimizers for the model itself, other than SGD (e.g Adam)? Perhaps using a different optimizer would make the learning rate scheduler unnecessary?
"However, post-processing calibration methods rely on an optimized independent output-probability mapping, which doesn’t alter the optimization process of the original model itself. Consequently, these methods can solely refine the probability distribution of the model output."- Why is this a disadvantage?
What are the computational requirements of the proposed method compared to other calibration methods- in particular does the tuning of the gradient decay coefficient greatly slow down optimization?

**Limitations:**

The authors have adequately addressed the limitations of their work.

---

> ### Author Rebuttal · Authors · 2024-08-04
>
> **Answers to Questions**
> * We appreciate the reviewer's technical comment. We had previously considered using the Adam optimizer instead of SGD to achieve a stable gradient. In response to your question, we show additional ablation studies to evaluate the optimization performance when Adam replaces SGD. Our experimental results indicate that Adam can indeed provide a more stable gradient and calibration performance, particularly in conjunction with our PID controller approach for model calibration. However, it is notable that Adam results in reduced accuracy, achieving only 63.5% on CIFAR-100 with ResNet35, significantly lower than the baseline accuracy of 73.8%.
>
> | SGD       | Adam      | PID Controller Approach | Gradient Compensation | Accuracy | ECE   | AdaECE |
> | --------- | --------- | ----------------------- | --------------------- | --------| ----- | ------ |
> | ✓         | -         | -                       | -                     | 73.8%   | 0.172 | 0.172  |
> | ✓         | -          | ✓                       | -                     | 72.5%   | 0.022 | 0.023  |
> | -         | ✓         | ✓                       | -                     | 63.5%   | 0.023 | 0.024  |
> | ✓         | -         | ✓                       | ✓                     | 74.7%   | 0.012 | 0.013  |
>
> A key difference arises in the baseline case handled by Adam. In our proposed PID controller method, which adjusts the hyperparameter $\beta$ during model calibration, the loss function is dynamic. While Adam retains previous gradient information, this can conflict with the current gradient vector direction because the optimization objective is dynamic. In contrast, our compensation method only modifies the learning rate and retains gradient vector direction pertinent to the current loss function. This may explain why the Adam optimizer does not yield better results.
>
> * Post-processing calibration methods necessitate the creation of an additional output-probability mapping $z \to p$ . Although this approach does not alter the decisions made in the classification task, it does increase the overall complexity of the model  $x \to z$. The input-probability mapping  $x \to p$ in post-processing calibration approaches is more complex than that in training-based calibration methods.
>
> * Our proposed method does not require significantly more computational resources compared to the baseline optimization strategy. It only involves adjusting the $\beta$ parameter in the loss function during the optimization process, and this adjustment is controlled by the PID of the RCE based on the validation set. Besides, through extensive empirical experiments, we find that the dynamic tuning mechanism of the gradient decay coefficient does not significantly impact the optimization of accuracy in the model. In other words, the dynamic gradient decay rate through PID control method for model calibration does not adversely affect model convergence regarding accuracy with proposed gradient compensation. Moreover, the PID parameter settings are not sensitive to the results, as shown in Figure 5.
>
> **Motivation for gradient decay coefficient**
>
> We would like to provide an additional explanation for reviewer to clarify the motivation of the gradient decay coefficient we have proposed.
>
> The probability-dependent gradient decay coefficient indicates the rate at which a sample's gradient magnitude decreases as the confidence of the softmax function increases (see Figure 1). A large gradient decay coefficient means that as the model's confidence increases, its gradient decreases more rapidly. Conversely, a small gradient decay coefficient implies that the gradient's magnitude decreases more slowly as confidence increases. This allows the sample to reach higher confidence levels at smaller gradient decay rates. As shown in Figure 2, Figures 6-9 in the Appendix, and Tables 4-6, the gradient decay rate $\beta$ exhibits a negative correlation with the confidence level for pairs of samples passing through the model. Higher gradient decay rates correlate with lower average confidence levels, motivating the use of a PID control method to manage sample confidence levels.
>
> Additionally, Equation (5) shows that the hyperparameter $\beta$ can be regarded as soft margin in the cross-entropy function with Softmax, allowing optimization of the samples to reach a soft confidence threshold. This approximation of the equation (5) on the max function demonstrates how this threshold compares across different class output. There is a compelling relationship between this hyperparameter $\beta$ and the model's confidence distribution. While this connection cannot be proven theoretically, it is supported by empirical experiments and logical reasoning. To the best of our knowledge, this is the first discussion of the probability-dependent gradient decay rate in relation to model calibration.
>
> Based on the above results, we adopt this probability-dependent gradient decay rate as the controlled variable of the controller for model calibration.

---

> > ### Comment · Reviewer_iW8b · 2024-08-08
> >
> > I would like to thank the authors for their detailed response. I have decided to leave my score unchanged.

---

### Author Rebuttal · Authors · 2024-08-04

Some supplementary Figures and Tables.

---

### Decision · Program_Chairs · 2024-09-25

**Decision:**

Accept (poster)

**Comment:**

This work proposes the use of a proportional-integral-derivative (PID) controller to address overconfidence in deep neural networks. Reviews were generally positive, commenting on the excellent presentation, the importance of addressing overconfidence and minimizing calibration error, and the surprising lack of PID in the literature given its practical success in industry. Initial concerns regarding novelty relative to competing methods, scalability, and sensitivity to minor alterations to methodology, were all addressed in the discussion phase. The authors included several additional experimental results in their rebuttal that are expected to greatly improve the work. I would also suggest that the authors increase the size of axis labels to ensure plots are easily visible in the final version. Subject to these inclusions, I recommend acceptance.